# A Dendritic Cell-Activating Rv1876 Protein Elicits Mycobacterium Bovis BCG-Prime Effect via Th1-Immune Response

**DOI:** 10.3390/biom11091306

**Published:** 2021-09-03

**Authors:** Seunga Choi, Han-Gyu Choi, Yong Woo Back, Hye-Soo Park, Kang-In Lee, Sintayehu Kebede Gurmessa, Thuy An Pham, Hwa-Jung Kim

**Affiliations:** 1Department of Microbiology, College of Medicine, Chungnam National University, Daejeon 35015, Korea; Seungachoi@cnu.ac.kr (S.C.); ekdrms20000@cnu.ac.kr (H.-G.C.); lenpk@nate.com (Y.W.B.); 01027192188@hanmail.net (H.-S.P.); popigletoh@nate.com (K.-I.L.); sintayehu.kebedeg@gmail.com (S.K.G.); anocyhp@gmail.com (T.A.P.); 2Department of Medical Science, College of Medicine, Chungnam National University, Daejeon 35015, Korea

**Keywords:** mycobacterium tuberculosis, dendritic cell, Rv1876, Th1 polarization, BCG-prime boost

## Abstract

The widely administered tuberculosis (TB) vaccine, Bacillus Calmette-Guerin (BCG), is the only licensed vaccine, but has highly variable efficiency against childhood and pulmonary TB. Therefore, the BCG prime-boost strategy is a rational solution for the development of new TB vaccines. Studies have shown that *Mycobacterium tuberculosis* (Mtb) culture filtrates contain proteins that have promising vaccine potential. In this study, Rv1876 bacterioferritin was identified from the culture filtrate fraction with strong immunoreactivity. Its immunobiological potential has not been reported previously. We found that recombinant Rv1876 protein induced dendritic cells’ (DCs) maturation by MAPK and NF-κB signaling activation, induced a T helper type 1 cell-immune response, and expanded the population of the effector/memory T cell. Boosting BCG with Rv1876 protein enhanced the BCG-primed Th1 immune response and reduced the bacterial load in the lung compared to those of BCG alone. Thus, Rv1876 is a good target for the prime-boost strategy.

## 1. Introduction

Tuberculosis (TB) remains a global health priority, with an estimated 2 billion people infected, according to the Global Tuberculosis Report, 2020. HIV infection, increase in immunosuppressive drug application, reactivation of latent TB, and the emergence of multidrug-resistant TB has complicated TB control. However, the only licensed vaccine, *Mycobacterium bovis* bacillus Calmette–Guérin (BCG), has highly variable efficiency in children, and is often ineffective in adults [1]. A safer and more effective vaccine to replace BCG or BCG-prime boosting is urgently required. After the World Health Organization (WHO) declared COVID-19 a pandemic, public health services, including TB services, were suspended in almost all countries. TB and COVID-19 are airborne diseases that primarily attack the lungs, but TB has a longer incubation period and a slower disease onset. The COVID-19 pandemic has impacted the treatment of TB patients in terms of increased diagnostic delays, fewer hospitalizations, and disruption of treatment services [2,3]. Due to this, the importance of vaccines to prevent TB infection has increased. Therefore, the identification and characterization of diverse mycobacterial components with vaccine potential are fundamental tasks to develop new vaccines against TB.

*Mycobacterium tuberculosis* (Mtb) is an intracellular pathogen that infects phagocytic antigen-presenting cells (APCs), including alveolar macrophages and dendritic cells (DCs) [4]. Although Mtb primarily infects alveolar macrophages leading to the pathogenesis of the disease, DCs play major roles in the regulation of cellular immune responses during Mtb infection [5,6]. The subunit vaccine based on antigen presentation for T cell priming could be an ideal target for BCG prime-boosting. Therefore, an understanding of the antimycobacterial mechanism of mycobacterial antigens that can mature and activate DCs can lead to the identification of an promising candidate for vaccine development.

DCs are crucial sentinel cells of the adaptive immune system [7,8]. In addition to processing and presenting antigens to activate naïve T lymphocytes, DCs determine whether it leads to tolerance or a type 1 or type 2 T cell response. These cells recognize mycobacterial antigens by expressing numerous pattern-recognition receptors (PRRs) such as Toll-like receptors that recognize motifs on pathogens, antigens, or substances [9,10]. Following antigenic stimuli, immature dendritic cells fully mature into APCs, upregulate surface co-stimulatory molecules and chemokines, secrete inflammatory cytokines [6,11], and migrate into lymphoid tissue presenting antigen-derived peptides associated with either class I or class II MHC molecules to naive CD8^+^ and CD4^+^ T cells [12,13]. Cytokines such as interleukin-12 (IL-12), IL-18, and tumor necrosis factor (TNF)-α, are produced by mature DCs and can induce T helper 1 (Th1)-cell responses [14]. In addition, although IL-12p40 promotes this migration to the draining node and initiates T cell activation, IL-10 may limit it [15,16,17]. IL-12 induces interferon-γ (IFN-γ) production and triggers CD4^+^ T cells to differentiate into Th1 cells in TB control [18,19,20]. Although IFN-γ and Th1 responses are relatively insufficient to control mycobacterial growth and protect against TB disease [21,22], many researchers have demonstrated their importance in the host immune response against Mtb [23,24].

This study identified an immune-reactive antigen from the multidimensional fractions of the complex Mtb antigen system, as described previously [25,26,27]. Rv1876 was identified from a fraction that strongly induced the activation of immune cells. Rv1876 is known as bacterioferritin, which is involved in storing iron to protect bacterial cells from iron overload; however, the immune response of immune cells has not been reported. This demonstrated that recombinant Rv1876 efficiently induces DC maturation and activation, leading to a protective Th1-immune response. Moreover, the Rv1876 protein showed a significant BCG-prime boosting effect in the Mtb infection mouse model. These results suggest that the DC-activating protein Rv1876 may be an promising candidate for TB vaccine development.

## 2. Materials and Methods

### 2.1. Animals

Specific pathogen-free 5- to 6-week-old female C57BL/6 mice and OT-II T-cell receptor (TCR) transgenic mice were purchased from the Jackson Laboratory (Bar Harbor, ME, USA). All mice were maintained at the Preclinical Research Center of Chungnam National University Hospital (Daejeon, Korea) and fed freely with sterile food and water under standardized light-controlled conditions (12-h light and 12-h dark periods).

### 2.2. Ethics Statement

The experiment was conducted in accordance with the Institutional Research Ethics Committee of Chungnam National University (approval number: 202003A-CNU-064) and the guidelines of the Food and Drug Administration.

### 2.3. Mycobacterial Strains and Medium

Mtb H37Rv (ATCC 27294) and H37Ra (ATCC 25177) were purchased from the American Type Culture Collection (ATCC, Manassas, VA, USA). *M. bovis* BCG (Tokyo strain) was provided by the Korean Institute of Tuberculosis (KIT). Mycobacteria were grown in Middlebrook 7H9 medium with 0.2% glycerol, 0.05% Tween-80 (Sigma, St. Louis, MO, USA), and 10% OADC (oleic acid, albumin, dextrose, and catalase; BD Biosciences, San Jose, CA, USA).

### 2.4. Abs and Reagents

All antibodies and cytokines used in this study are described in the Appendix A (Appendix A). Reagents were purchased from PeproTech (Rocky Hill, NJ, USA), CreaGene (Gyeonggi, Republic of Korea), BD Biosciences (San Jose, CA, USA), InvivoGen (San Diego, CA, USA), Cell Signaling Technology (Danvers, MA, USA), Merck (Darmstadt, Germany), and Thermo Fisher Scientific (Waltham, MA, USA).

### 2.5. Multi-Dimensional Fractionation of Mtb Culture Filtrate Proteins (CFPs)

Mtb H37Rv was grown for 6 weeks at 37 °C as surface pellicles in Sauton’s medium, and then the CFPs were prepared as previously described [27]. Briefly, Mtb CFPs were fractionated using multistep chromatography. An 80% ammonium sulfate precipitate of the CFPs was fractionated into seven fractions by hydrophobic interaction chromatography (HIC), and further fractionated by hydroxyapatite chromatography (HAT) and ion-exchange chromatography (IEC). IEC fraction number 70 of the HAT pass fraction from initial fraction 5 was further fractionated by whole gel eluter. The major single band of gel eluter fraction 3 was identified as Rv1876 by liquid chromatography-electrospray ionization-tandem mass spectrometry (LC-ESI/MS).

### 2.6. Construction and Production of Recombinant Rv1876 Protein

To obtain the sequence of Rv1876, the DNA fragment was amplified by PCR from purified M. tuberculosis H37Rv genomic DNA and the primers Rv1876 forward, 5′-CATATGCAAGGTGATCCCGATGTTCTG-3′, and reverse, 5′-AAGCTTGGTCGGTGGGCGAGAGACGCA-3′. The DNA fragment of Rv1876 was cloned into the *NdeI* and *HindIII* sites of the plasmid pET22b (+) vector (Novagen, Madison, WI, USA), and the c-terminal His-tag present in the plasmid was included.

The product was transformed into *E. coli* BL21, which was induced to express proteins by isopropyl β-D-1-thiogalactopyranoside (IPTG). IPTG was added at a final concentration of 1 mM. Finally, Western blotting confirmed the expression of Rv1876. The recombinant protein was prepared as described previously [27]. The amount of residual LPS in the Rv1876 protein preparation was evaluated using a LAL test kit (Lonza, Basel, Switzerland), according to the manufacturer’s instructions.

### 2.7. Bone Marrow Cells Isolation and Culture

The tibias and femurs were separated from C57BL/6 mice. Both ends of the bone were cut off, and a 26G needle of a syringe was inserted into the bone cavity to rinse with the medium. The cells were collected and centrifuged at 1500 rpm for 5 min, and the supernatant was discarded. The pelleted cells were resuspended in red blood cell lysis buffer to lyse RBCs. Following the second centrifugation, the cells were collected, and murine bone marrow-derived DCs (BMDCs) and bone marrow-derived macrophages (BMDMs) were generated, cultured, and purified, as recently described [28].

To differentiate into BMDCs, the BM-isolated cells were plated in six-well culture plates (1 × 10^6^ cells/mL) and cultured at 37 °C in an incubator containing 5% CO_2_. The cells were suspended in RPMI 1640 medium supplemented with 100 unit/mL penicillin/streptomycin (Lonza), 10% fetal bovine serum (Lonza), 50 μM mercaptoethanol (Lonza), 0.1 mM non-essential amino acids (Lonza), 1 mM sodium pyruvate (Sigma), 20 ng/mL GM-CSF, and 10 ng/mL IL-4, respectively.

To differentiate into BMDMs, the BM-isolated cells were suspended in Dulbecco’s modified Eagle’s medium (DMEM) supplemented with 100 unit/mL penicillin/streptomycin (Lonza) and 10% fetal bovine serum (Lonza). The cells were cultured for 6 days with 20 ng/mL M-CSF. The medium was replaced after 3 days of incubation. The adherent cells were detached by trypsin (0.5%) digestion and harvested by centrifugation.

### 2.8. Analysis of the Surface Molecules by FACS

On day 7, for dendritic cell activation, the cells were stimulated with Rv1876 protein or LPS for 24 h. Activated BMDCs were harvested and resuspended in FACS buffer (PBS, 2% FBS and 0.1% NaN_3_ sodium azide). FACS staining was performed with PE-conjugated antibodies against surface molecules for 30 min at 4 °C. The stained cells were washed with FACS washing buffer and fixed with 0.4% paraformaldehyde in PBS. Fluorescence was measured using a NovoCyte flow cytometer (ACEA Biosciences, San Diego, CA, USA). The data were analyzed using Novoexpress software and FlowJo software (Tree Star, Ashland, OR, USA).

### 2.9. Immunoblotting Analysis

After stimulation, the cells were washed with ice-cold PBS and collected by centrifugation. Cell pellets were lysed in lysis buffer containing 50 mM Tris-HCl (pH 7.5), 150 mM NaCl, 1% Triton X-100, 1 mM EDTA, 1 mM phenylmethanesulfonyl fluoride, and protease inhibitor cocktail (Sigma).

Whole-cell lysate samples were separated on SDS-polyacrylamide gels and then transferred to a polyvinylidene difluoride (PVDF) membrane. The membrane was blocked in 5% nonfat dried milk in TBST (20 mM Tris-HCl [pH 7.4], 137 mM NaCl, and 0.1% Tween 20) and incubated with a primary antibody overnight at 4 °C. After washing with TBST, the membranes were incubated with HRP-conjugated secondary antibodies for 2 h. Furthermore, after washing in TBST, the membrane was developed using the ECL system (Millipore, MA, USA) following the manufacturer’s instructions.

### 2.10. Immunofluorescence Staining

The cells on the coverslips were washed in PBS and fixed in 4% paraformaldehyde (PFA) for 20 min at room temperature. The fixed cells were washed and permeabilized by incubation in PBS containing 0.1% Triton X-100 (Sigma). The fixed cells were incubated with the diluted primary antibody in PBS (1% BSA) overnight at 4 °C. The cells were then washed and incubated in diluted secondary antibody for 2 h. After washing in PBS, the coverslips were mounted onto glass slides. Images were acquired using a super resolution confocal laser scanning microscope controlled by Las X software (Leica Microsystems, Wetzlar, Germany).

### 2.11. Analysis of Signaling Pathway in DCs Using Pharmacological Inhibitors

All the pharmacological inhibitors were purchased from Calbiochem. The cells were treated with an inhibitor for 60 min, followed by treatment with protein for 24 h. The concentration information and manufacturing of pharmacological inhibitors has been previously described [23].

### 2.12. An In Vitro T-Cell Proliferation Assay

The spleen of OT-II TCR Tg mice was ground in homogenizers, and the resultant homogenate was passed through a nylon mesh into a 50 mL conical-bottom tube. OVA-specific CD4^+^ T cells were isolated using a MACS column (Miltenyi Biotec, Auburn, CA, USA) according to the manufacturer’s protocol. Purified CD4^+^ T cells were stained with 1 μM CFSE (Invitrogen), as previously described [27]. DCs were treated with the synthetic OVA_323–339_ peptide (Invivogen) in the presence of protein for 24 h. The CFSE-stained CD4^+^ T cells (5 × 10^4^ cells/well) were cultured with protein-stimulated DCs (5 × 10^3^ cells/well) in 96-well culture plates at a DC:T cell ratio of 1:10. On day 3 of co-culture, each batch was stained with APC-eFluor 780 conjugated anti-CD4^+^ mAb and analyzed using flow cytometry.

### 2.13. Analysis of the Expansion of Effector/Memory T Cells Using FACS

As previously mentioned, CD4^+^ T cells, which participate in coculture, were isolated using a MACS column (Miltenyi Biotec) from the splenocytes of Mtb-infected mice. DCs (2 × 10^5^ cells per well) were treated with Rv1876 for 24 h, followed by extensive washing, and were cocultured with 2 × 10^6^ CD4^+^ T cells (BCG-infected T cells) at DC:T cell ratios of 1:10. After 4 days of coculture, the harvested cells were stained with BV605-conjugated anti-CD4^+^ mAb, FITC-conjugated anti-CD62L mAb, and PE-conjugated anti-CD44 mAb, then analyzed using flow cytometry. The expression of cytokines in the supernatants was measured using a commercial sandwich enzyme-linked immunosorbent assay (ELISA) kit (Invitrogen) according to the manufacturer’s instructions.

### 2.14. Cytokine Measurements

The production of IL-1β, TNF-α, IFN-γ, IL-4, IL-2, IL-12p70, and IL-23p19 by cells was measured in culture supernatants using specific ELISA kits (Invitrogen) following the manufacturer’s instructions.

### 2.15. Measurement of Intracellular Mtb Growth in BMDMs

The BMDMs were seeded at 2 × 10^5^ cells/well and infected with Mtb at a MOI of 1 for 4 h. The infected BMDMs were incubated with 200 μg/mL amikacin for 2 h and washed with PBS to eliminate extracellular Mtb. The T cell mixture of CD4^+^ T-cells previously cocultured for 3 days with protein-activated DCs was added to the infected BMDMs in each well and further incubated for 3 days. The supernatant was collected, and the cells were lysed in sterile distilled water for 30 min at 37 °C. The number of internalized Mtb cells in the BMDMs was calculated by lysing the cells from each well. The cell lysates were serially diluted and plated on Middlebrook 7H10 agar (BD Biosciences) supplemented with 0.05% glycerol, amphotericin B (Sigma-Aldrich, St. Louis, MO, USA), and 10% OADC at 37 °C.

### 2.16. Vaccination and Challenge of Mice

For vaccination, female C57BL/6 mice, aged 6 weeks, were subcutaneously injected with BCG (2 × 10^5^ CFUs/mouse). Twelve weeks post-BCG immunization, Rv1876 protein (5 μg) was administered three times at 3-week intervals with dimethyldioctadecylammonium liposomes (DDA/250 µg)-monophosphoryl lipid-A (MPL/25 µg) purchased from Sigma-Aldrich. The vaccination mixture was manufactured according to a previously described procedure [29].

Immunized mice were intratracheally infected with Mtb four weeks after the last immunization to verify the protective effect. Each mouse was inoculated with 1 × 10^6^ CFU of Mtb H37Ra in 50 µL saline, as previously described [30]. Mtb infection was performed in a B2 type Biological Safety Cabinet Class II (ESCO, Seoul, Korea). Lung cells were harvested from each group at four weeks after Mtb infection, and the population of antigen-specific T cells from the lung was assessed using flow cytometry.

### 2.17. Intracellular Cytokine Assays Using FACS

Harvested lung cells were dissociated using a lung dissociation kit (MACS) following the manufacturer’s instructions to generate single-cell suspensions. Single-cell suspensions were stimulated with protein for 24 h at 37 °C in the presence of GolgiStop (BD Biosciences) according to a previously described procedure [28]. The stained cells were fixed and permeabilized by a Cytofix/Cytoperm Kit (BD Biosciences) according to the manufacturer’s instructions. Intracellular cytokines were detected using fluorophore-conjugated antibodies against cytokines. Stained cells were analyzed using flow cytometry.

### 2.18. Measurement of Bacterial Counts

The lungs and spleens from mice in each group were homogenized. The homogenized cell lysates were serially diluted, and the number of viable bacteria was determined by plating onto Middlebrook 7H10 agar supplemented with 10% OADC, 0.25% amphotericin B, and 0.05% glycerol. Colonies on plates were counted after 3 or 4 weeks of incubation at 37 °C. Data on the CFUs are reported as the median log_10_ CFU ± interquartile range (IQR).

### 2.19. Statistical Analysis

The experiments were performed at least thrice, with consistent results. The levels of significant differences between groups were determined by a Tukey’s multiple comparison test distribution using GraphPad Prism (GraphPad Software, San Diego, CA, USA). The data in the graphs are expressed as the mean ± SEM. Differences with each value of * *p* < 0.05, ** *p* < 0.01, or *** *p* < 0.001 were considered statistically significant.

## 3. Results

### 3.1. Rv1876 Identified from Mtb Culture Filtrates Induces DC Maturation and Activation

Mtb CFPs were fractionated using multistep chromatography, as described in the Material and Methods section (Figure 1Aa and Appendix A). IEC fraction number 70 of the HAT pass fraction from the initial fraction 5 was further fractionated by the whole gel eluter (Figure 1Ab). Each fraction was verified for its ability to secrete pro-inflammatory cytokines from the immune cells (data not shown). The primary single band of gel eluter fraction 3 was identified as Rv1876 by LC-ESI/MS.

To investigate the immunoactivity of Rv1876 protein on DCs, Rv1876 protein was produced in *E. coli*. The purified recombinant protein showed a single band at 20 kDa on SDS-PAGE and was detected using anti-His antibody (Figure 1Ac). The purified protein lots with extremely low endotoxin content were used in subsequent experiments (<0.07 EU/mL, data not shown). DCs treated with Rv1876 displayed dose-dependent enhanced expression of MHC class I and II and costimulatory molecules such as CD80 and CD86 (Figure 1B). Additionally, they produced significantly higher amounts of IL-1β, TNF-α, IL-12p70, and IL-23p19 than untreated cells (Figure 1C). LPS was used as the positive control. Furthermore, to assess LPS contamination using polymyxin B treatment, we confirmed that cytokine production by the Rv1876 protein was not inhibited by polymyxin B treatment, while that of LPS was significantly inhibited. (Appendix A). These results suggest that Rv1876 causes DCs to secrete proinflammatory cytokines.

Thereafter, we tested whether MAPKs and NF-κB, which regulate DC maturation and inflammatory responses [31], are involved in Rv1876-mediated DC activation. We verified that the phosphorylation of p38, ERK1/2, ERK, and the phosphorylation and degradation of IκB-α were induced by Rv1876 treatment. (Figure 1D). DCs treated with Rv1876 showed translocation of p65 into the nucleus by the degradation of IkB-α (Appendix A). In addition, the expression of costimulatory molecules and inflammatory cytokines by Rv1876 treatment was significantly suppressed by pre-treatment with a p38 inhibitor, an ERK1/2 inhibitor, a JNK inhibitor, or an NF-κB inhibitor (Figure 1E,F). Through these results, we found that the MAPK and NF-κB signaling pathways are required for DC maturation and activation by the Rv1876 protein.

### 3.2. Rv1876-Matured DCs Induce CD4^+^ T-Cell Proliferation and Expansion of the Effector/Memory T-Cell Population

To investigate the interaction between Rv1876 protein-activated DCs and CD4^+^ T cells, we examined whether Rv1876 protein-activated DCs would induce CD4^+^ T cell proliferation using OT-II T cell receptor (TCR) transgenic CD4^+^ T cells. A syntergenic in vitro T cell proliferation assay in which CFSE-labeled OVA-specific CD4^+^ T cells were stimulated with DCs pulsed with OVA peptide for 72 h was conducted. Rv1876-, Ag85-, or LPS-matured DCs induced the proliferation of CD4^+^ T cells to a significant extent compared to those induced by untreated DCs (Figure 2A). Thereafter, we examined whether CD4^+^ T cells from BCG-infected mice could be efficiently activated by Rv1876-matured DCs. Ag85, which is known as a strong T-stimulating antigen, was used as a control mycobacterial antigen. CD62L downregulation and CD44 upregulation in syngeneic CD4^+^ T cells co-cultured with Rv1876-matured DCs were significant compared to those in the control and LPS- or Ag85-treated DCs (Figure 2B,C). Under the same conditions, the secretion of IFN-γ or IL-2 was significantly higher in T cells co-cultured with Rv1876-matured DCs compared to that in the untreated or LPS-stimulated DCs (Figure 2D). IL-2 production was significantly higher in T cells activated with Rv1876-matured DCs than in Ag85-matured DCs. Additionally, the secretion of IL-4 was not significantly different in CD4^+^ T cells incubated with antigen-treated DCs. These findings indicate that Rv1876-matured DCs induce a Th1 cell-immune response and expand the population of effector/memory T cells.

### 3.3. T Cells Activated by Rv1876-Matured DCs Play a Role in the Inhibition of Intracellular Mtb Growth

Next, we determined that T cells activated by Rv1876-matured DCs could control intracellular Mtb growth. Purified T cells from the spleen of BCG-infected mice were activated by co-culture with Rv1876-matured DCs for 3 days. Then, the T cells were added to Mtb-infected BMDMs and further cultured for 3 days. As shown in Figure 3A, the addition of T cells activated by LPS- or Ag85-stimulated DCs appeared to slightly inhibit intracellular Mtb growth. Notably, T cells activated by Rv1876-mature DCs significantly inhibited Mtb growth compared to untreated, LPS, or Ag85-mature DCs, and also induced significantly higher NO and IFN-γ production in the culture supernatants when compared with other activated T cells; Rv1876-matured DCs activated T cells to secrete higher levels of NO and IFN-γ, which are associated with antimycobacterial activity (Figure 3B,C). These results demonstrated that Rv1876-matured DCs induced the activation of T cells to improve their bactericidal activity.

### 3.4. The Rv1876 Protein Enhances the BCG Protective Boosting Effects against Mtb

Furthermore, we measured the protective efficacy of Rv1876 as a BCG-prime booster in a mouse infection model. The mice were immunized three times with Rv1876/DDA-MPL or DDA-MPL after BCG injection, as shown in Figure 4A. Four weeks after the final vaccination, the mice were infected with Mtb, and the bacterial loads in the lungs were measured 4 weeks post-challenge. The lungs of the BCG-DDA-MPL and BCG-Rv1876/DDA-MPL groups showed significantly reduced bacterial loads compared to those of the infection control group (Figure 4B). In particular, a significantly lower bacterial count was observed in the lungs of the BCG-Rv1876/DDA-MPL group than in the BCG-DDA-MPL group. The expression of cytokines in lung cells before challenge and 4 weeks after Mtb challenge was stimulated ex vivo with Rv1876. The secretion of cytokines from the culture supernatant was measured using ELISA (Figure 4C). The production of IL-2 and IFN-γ was significantly higher before and after challenge in lung cells from Rv1876-boosted mice than in mice immunized with BCG alone. These findings indicate that boosting BCG with Rv1876 protein enhances the protective effect and elicits a Th1 immune response in a mouse model.

### 3.5. Multifunctional T-Cells in Mice Boosting with Rv1876 Correlate to Protection Efficacy

Thereafter, we evaluated the generation of antigen-specific multifunctional T cells that produce IFN-γ, TNF-α, IFN-γ, and IL-2 in lung cells upon ex vivo re-stimulation with Rv1876 before and after Mtb challenge. Antigen-stimulated CD4^+^ T cells were stained for multiple intracellular cytokines and analyzed using flow cytometry. A significant increase in antigen-specific CD4^+^CD44^+^IFN-γ^+^IL-2^+^ T cells before challenge and CD4^+^CD44^+^IFN-γ^+^TNF-α^+^IL-2^+^ T cells/IFN-γ^+^IL-2^+^ T cells were induced in the lung cells of mice boosted with Rv1876, compared to that of mice injected with only BCG. Thereafter, we determined whether these multifunctional T cell expansions were correlated with protective efficacy. Therefore, the relationship between the number of expanded multifunctional T cells and the bacterial load in the lungs of individual mice was measured. As shown in Figure 5B, the number of CD4^+^CD44^+^IFN-γ^+^TNF-α^+^IL-2^+^ T cells was inversely correlated with bacterial loads before (R = −0.8404, *p* < 0.0001) and after challenge (R = −0.8393, *p* < 0.0001). In addition, there were correlations between CD4^+^CD44^+^IFN-γ^+^IL-2^+^ T cells and CD4^+^CD44^+^TNF-α^+^IL-2^+^ T cells and protective efficacy (Appendix A). These results suggest that multifunctional T cells induced by Rv1876 boosting were effectively re-expanded by Mtb infection and played an essential role in clearing the bacteria.

## 4. Discussion

Iron is a crucial micronutrient for mycobacterial growth [32,33], and Mtb limits the cellular levels of free iron through various storage mechanisms [34]. The delivery of iron, performed by a superfamily of proteins known as ferritins, regulates the biosynthetic functions of the cell. These proteins play a role in iron storage and uptake regulation, and they could contribute to Mtb growth, virulence, latency, and aminoglycoside drug resistance [35,36,37,38,39]. Previous studies have reported that the expression of bacterioferritin (Rv1876) and ferritin (Rv3841) increases in aminoglycoside-resistant clinical isolates, which suggests that they are a factor in resistance [36,37]. The Rv1876 protein was previously identified in the membrane and culture filtrate of Mtb H37Rv [40,41,42]. Bacteriophage ferritin is considered an important virulence factor that regulates iron deficiency and stores excess iron to protect cells from oxidative stress or iron overload [43]. Many researchers have used virulence factors as subunit vaccines [44]. Previous studies have revealed the role of secreted antigens in enhancing protective immunity through DC maturation [45]. For example, Mtb protein Rv0577 is present in culture filtrates as a prominent antigen in TB patients [46]. Also, Rv0577 protein has been reported to be an agonist of toll-like receptor 2 (TLR-2), inducing DC maturation and driving a Th1 immune response [47]. We have investigated secreted antigens capable of stimulating immunoreactivity through multidimensional fractionation of Mtb CFPs [25,26,27]. Rv1876, an immunostimulatory antigen, has not been previously known for its immunobiological potential despite being an essential protein for mycobacterial growth.

Protective immunity against TB is influenced by CD4 and CD8 T cell functions [48]. In various TB infection models, several groups have shown that the activation of a T cell-mediated immune response requires the selection of effective vaccine candidates. Recently, we found that Mtb ferritin antigens (Rv3841) are capable of eliciting dominant Th1 responses associated with reduced bacterial burden in Mtb-infected macrophages [49]. Here, we found immunoreactivity of the bacterioferritin Rv1876, which belongs to the family of Mtb proteins associated with virulence. Recombinant proteins require adjuvants to drive successful adaptive immune responses due to their poor immunogenicity. We used monophosphoryl lipid A (MPL-DDA), which has been successfully combined with antigen proteins to induce protective immune responses against TB. This study characterized the immune responses induced by Rv1876/MPL-DDA in mice, and we report the BCG prime-boost efficacy of this vaccine candidate against Mtb.

The results of this study indicate that recombination of Rv1876 protein induces DC maturation via the activation of MAPK and NF-κB signaling. Furthermore, we found that DCs matured by the Rv1876 protein induced T cell immune responses toward Th1 polarization. DCs matured by Rv1876 significantly expanded the population of CD44^high^CD62L^low^ CD4^+^ effector/memory cells from splenic T cells from Mtb-infected mice. The results clearly showed that DCs matured by Rv1876 induced the expansion of effector/memory T cells with high expression of immunostimulatory cytokines, significantly polarized towards the Th1 phenotype. Based on our results, we verified that Rv1876, the DC-activating antigen, can induce a Th1 memory response as a specific recall antigen, which implies the possibility of being a candidate for vaccine design.

The prime-boost strategy that combines BCG with a subsequently administered vaccine candidate is currently one of the preferred approaches to TB vaccine development. A faster increase in human life expectancy will lead to faster population aging. Consequently, TB in the elderly has become an increasingly global health problem. According to previous reports, the elderly with tuberculosis are a vulnerable group that is difficult to diagnose and has a high mortality rate [50]. Therefore, the necessity of developing a BCG booster vaccine for adults is growing. Several studies have shown that BCG protection can be improved using subunits or viral-based vaccines [51]. However, few studies have focused on boosting BCG using DC-activating antigens. Although the interaction of DC-mycobacterial antigens plays an essential role in the protective immune system against microbial infection, DC-activating antigens have never been considered a target of boost BCG vaccines.

Many people worldwide have been vaccinated with BCG; however, prophylactic efficacy is not maintained throughout life. Therefore, boosting protective immunity by BCG is considered an essential development towards a new TB vaccine. However, homologous boosting with BCG was ineffective in humans [52], and severe lesions were found in guinea pigs [53]. Therefore, heterologous strategies, priming with BCG and boosting with adjuvanted [54,55] or vectored [56] mycobacterial antigens, have been considered promising. The Mtb antigen Rv1876 is also found in BCG; thus, Rv1876 may be an effective vaccine candidate for enhancing BCG-induced immunity.

This study demonstrated that boosting BCG with Rv1876 protein enhanced the BCG-primed Th1 immune response and reduced lung CFU compared to the protective effect induced by BCG alone. Our findings also showed an increase in the number of triple-positive IL-2^+^TNF-α^+^IFN-γ^+^ antigen-specific CD4^+^ T cells from the lungs of mice boosted BCG with Rv1876 for protection against Mtb infection. Although an increase in multifunctional CD4^+^ T cells after boosting BCG with MVA85A did not improve protection [57], several studies have shown that the expansion of the multifunctional T cell population is closely related to protection against TB [58,59,60,61], agreeing with our results.

Although our study had some limitations, we speculated that Rv1876, a DC-activating antigen, could be a good target for the prime-boost strategy. To prove that enhancing the boost effect of BCG with Rv1876 protein is a promising strategy, further studies on protection against high-virulence mycobacterial strains should be conducted.

## 5. Conclusions

The development of a BCG-booster vaccine could help reduce financial burdens on the public health system and help control the disease. Extensive knowledge of vaccine candidates gained from mycobacterial antigen studies can shorten the vaccine development process. The results of this study show that Rv1876, a DC-activating antigen, can induce a Th1 memory response as a specific recall antigen, suggesting that it is an effective vaccine candidate for enhancing protective immunity. This study demonstrated that boosting BCG with Rv1876 protein enhanced the BCG-primed Th1 immune response and reduced lung CFU compared to the protective effect induced by BCG alone. Therefore, we speculate that Rv1876 could be a good target for the prime-boost strategy.

## Figures and Tables

**Figure 1 biomolecules-11-01306-f001:**
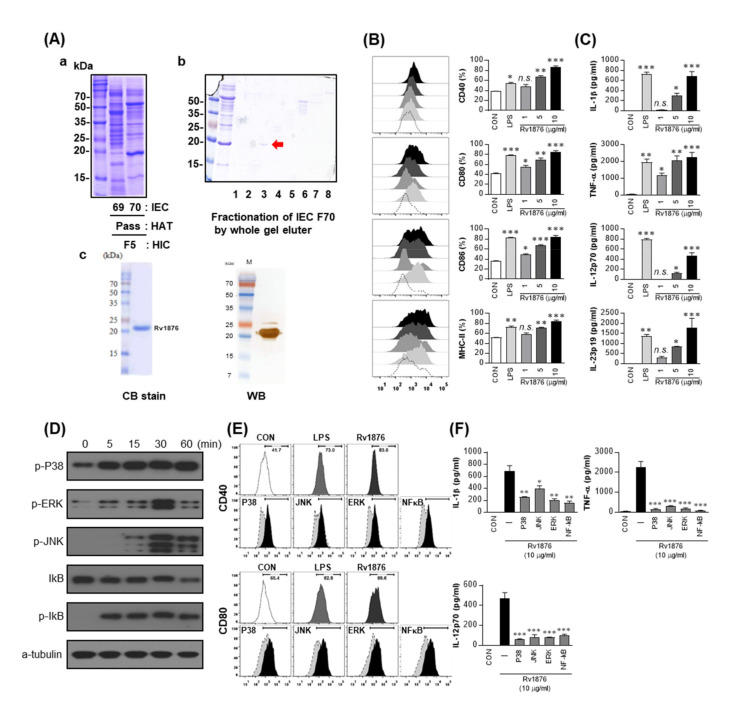
Preparation of Rv1876 and induction of DC maturation by Rv1876. (**A**) (**a**) The 80% ammonium sulfate precipitate of the CFPs was fractionated by hydrophobic interaction chromatography (HIC) using Phenyl Sepharose. The primary fractions were divided and concentrated into seven fractions. Each of the primary fractions was further fractionated using hydroxyapatite chromatography (HAT) using stepwise elution with NaCl. A third fractionation was performed by DEAE ion-exchange chromatography using gradient elution. The eluates were pooled into 80 fractions based on protein band pattern, and were concentrated. (**b**) Finally, the interesting fractions were further fractionated using whole gel eluter. The proteins were analyzed using SDS-PAGE with Coomassie brilliant blue. (**c**) Recombinant Rv1876 produced in BL21 cells was purified using an NTA resin. The purified protein was subjected to SDS-PAGE and Western blot analysis using a mouse anti-His Ab. (**B**) Rv1876 induced phenotypic and functional activation of DCs in a dose-dependent manner. Immature DCs (1 × 10^6^ cells/mL) were cultured in the presence of GM-CSF and IL-4 alone (control) or GM-CSF, IL-4 or 1, 5 or 10 μg/mL Rv1876 or GM-CSF, IL-4 or 100 ng/mL LPS for 24 h and were analyzed for the expression of surface markers using flow cytometry. The cells were gated on CD11c^+^. The DCs were stained with anti-CD40, anti-CD80, anti-CD86, or anti-MHC class II. The percentage of positive cells is shown in each panel. The bar graphs depict the mean values ± SD (*n* = 3). The levels of significance (* *p* < 0.05, ** *p* < 0.01 or *** *p* < 0.001, determined by one-way ANOVA test) of the differences between the treatment data and the control data are indicated. Treatments with no significant effect are indicated by n.s. (**C**) DCs were generated by stimulating immature DCs with 100 ng/mL LPS or 1, 5, or 10 μg/mL Rv1876 for 24 h. The quantities of TNF-α, IL-1β, IL-23p19, and IL-12p70 in the culture supernatant were determined using an ELISA. All data were expressed as the mean values ± SD (*n* = 3). The levels of significance (* *p* < 0.05, ** *p* < 0.01 or *** *p* < 0.001 determined by one-way ANOVA test) of the differences between the treatment data and the control data are indicated; treatments that were not significantly different are indicated by *n.s.* (**D**) Protein production over time by DCs treated with 10 μg/mL Rv1876. Cell lysates were subjected to SDS-PAGE, and immunoblotting analysis was performed using Abs specific to phospho-p38 (p-p38), phospho-ERK1/2 (p-ERK1/2), phospho-JNK (p-JNK), phosphor-IκB-α (p-IκB-α), and IκB-α. α-tubulin was used as the loading control for the cytosolic fractions. Representative blots out of five independent experiments are shown. (**E**,**F**) DCs were treated with pharmacological inhibitors of p38 (SB203580, 20 μM), ERK1/2 (U0126, 10 μM), JNK (SP600125, 20 μM), and NF-κB (Bay11-7082, 20 μM) for 1 h before the treatment with 10 μg/mL Rv1876 for 24 h. (**E**) The expression of co-stimulatory molecules was determined using flow cytometry. (**F**) The amounts of TNF-α, IL-1β, and IL-12p70 in the culture media were determined using an ELISA. Untreated by inhibitor indicated (-). Mean values ± SD (*n* = 3) are shown; * *p* < 0.05, ** *p* < 0.01 or *** *p* < 0.001 = significant vs Rv1876-treated DCs determined using unpaired Student’s *t*-test.

**Figure 2 biomolecules-11-01306-f002:**
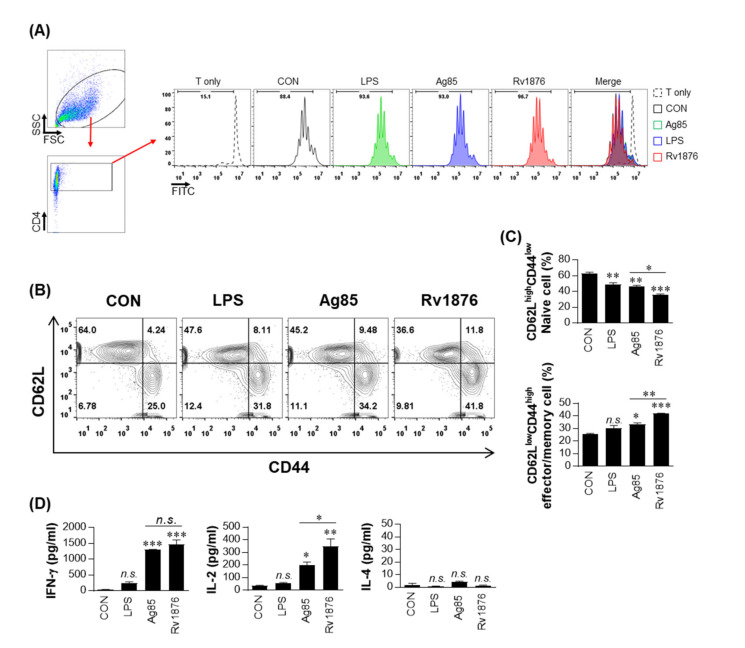
Rv1876-treated DCs stimulate T-cells to produce Th1 cytokines and induce the expansion of the effector/memory T-cell population. (**A**,**B**) Transgenic OVA-specific CD4^+^ T cells were isolated using MACS from OT-II mice splenocytes, stained with CFSE, and co-cultured for 96 h with DCs that had been treated with Rv1876 (10 μg/mL), Ag85 (10 μg/mL), or LPS (100 ng/mL) and pulsed with OVA_323–339_ (1 μg/mL) to produce OVA-specific CD4^+^ T cells. (**A**) The proliferation of OT-II T-cells was then assessed using flow cytometry. T cells alone and T cells co-cultured with untreated DCs served as the controls. Representative histograms from three independent experiments are shown. (**B**–**D**) DCs were treated with Rv1876 (10 μg/mL) or LPS (100 ng/mL) and cocultured for 3 days with T cells from BCG-infected mice at DC to T cell ratios of 1:10. Splenocytes were stained with anti-CD4, anti-CD62L, and anti-CD44 mAbs. (**B**) Bar graphs show CD62L^low^CD44^high^ T cells or CD62L^high^CD44^low^ T cell populations among the spleen cells. (**C**) A histogram is shown for the gating of the labeled T cells. All data are expressed as the mean ± standard deviation (SD); * *p* < 0.05, * *p* < 0.01, or *** *p* < 0.001 for treated samples compared to non-treated DCs co-cultured with CD4^+^ T cells; n.s, not significant, as determined by one-way ANOVA. (**D**) Culture supernatants were harvested after 96 h, and the IFN-γ, IL-2, and IL-4 levels were determined using ELISAs. All data are expressed as the mean ± standard deviation (SD); * *p* < 0.05, * *p* < 0.01, or *** *p* < 0.001 for treated samples compared to non-treated DCs co-cultured with CD4^+^ T cells; n.s., not significant, as determined by one-way ANOVA.

**Figure 3 biomolecules-11-01306-f003:**
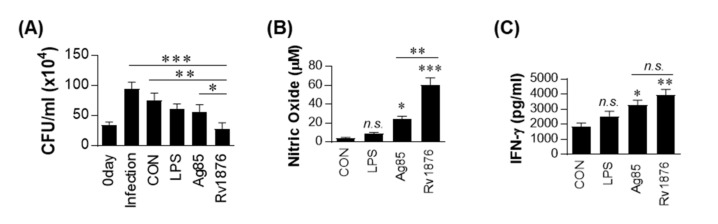
T cells activated by Rv1876-maturated DCs inhibit intracellular Mtb growth. Ag85 or Rv1876-stimulated DCs at a DC:T cell ratio of 1:10 for 3 d were cocultured with BMDMs infected with Mtb. (**A**) Intracellular Mtb growth in BMDMs was determined at time point 0 (0 days) and 3 d after coculturing with T cells or without T cells (control). All data are expressed as the mean ± standard deviation (SD); * *p* < 0.05, ** *p* < 0.01, or *** *p* < 0.001 for treated samples between infection control vs Rv1876-treated DCs co-cultured with CD4^+^ T cells, non-treated DCs co-cultured with CD4^+^ T cells vs Rv1876-treated DCs co-cultured with CD4^+^ T cells, and Ag85-treated DCs co-cultured with CD4^+^ T cells vs Rv1876-treated DCs co-cultured with CD4^+^ T cells, as determined by one-way ANOVA. (**B**) The NO production in culture supernatants. All data are expressed as the mean ± standard deviation (SD); * *p* < 0.05 or *** *p* < 0.001 for treated samples compared to non-treated DCs co-cultured with CD4^+^ T cells, ** *p* < 0.01 for treated samples compared to Ag85-treated DCs co-cultured with CD4^+^ T cells vs Rv1876-treated DCs co-cultured with CD4^+^ T cells; n.s., not significant, as determined by one-way ANOVA. (**C**) IFN-γ levels in culture supernatants were measured by ELISA. All data are expressed as the mean ± standard deviation (SD); * *p* < 0.05 or ** *p* < 0.01 for treated samples compared to non-treated DCs co-cultured with CD4^+^ T cells, and Ag85-treated DCs co-cultured with CD4^+^ T cells vs Rv1876-treated DCs co-cultured with CD4^+^ T cells; n.s., not significant, as determined by one-way ANOVA.

**Figure 4 biomolecules-11-01306-f004:**
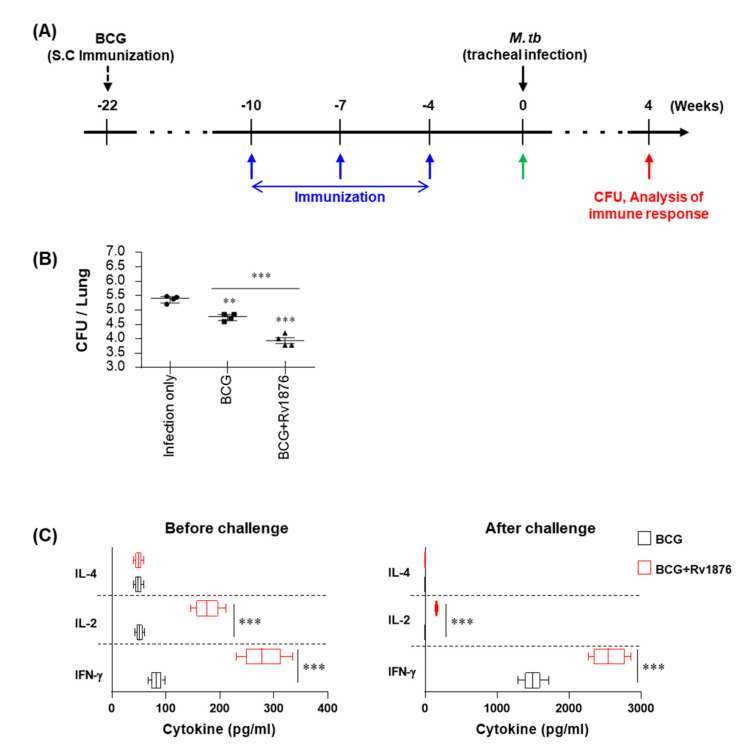
Rv1876/MPL-DDA booster vaccination improves BCG-primed protection against Mtb. (**A**) Experimental design for Rv1876 subunit vaccine testing. Mice (*n* = 12 per group) were immunized by BCG injection 12 weeks before subunit vaccination. MPL-DDA were administered (blue arrows) before Mtb challenge (black arrow). Immunological analysis was conducted before (green arrow) and after Mtb infection (red arrow). Bacterial counts in each group were determined at the indicated time points after Mtb infection (red arrow). (**B**) CFUs in the lungs in all treatment groups at 4 weeks post-infection, determined by counting the viable bacteria. Data are from one of two independent experiments (*n* = 4 mice per group at each time point). Mann–Whitney rank tests were used to compare groups. ** *p* < 0.01, and *** *p* < 0.001. (**C**) Levels of IFN-γ, IL-2, and IL-4 secreted by lung cells from each fully immunized group in response to Rv1876 (2 μg/mL) stimulation as detected by ELISA. *** *p* < 0.001 compared to BCG-immunized mice determined by unpaired Student’s *t*-test.

**Figure 5 biomolecules-11-01306-f005:**
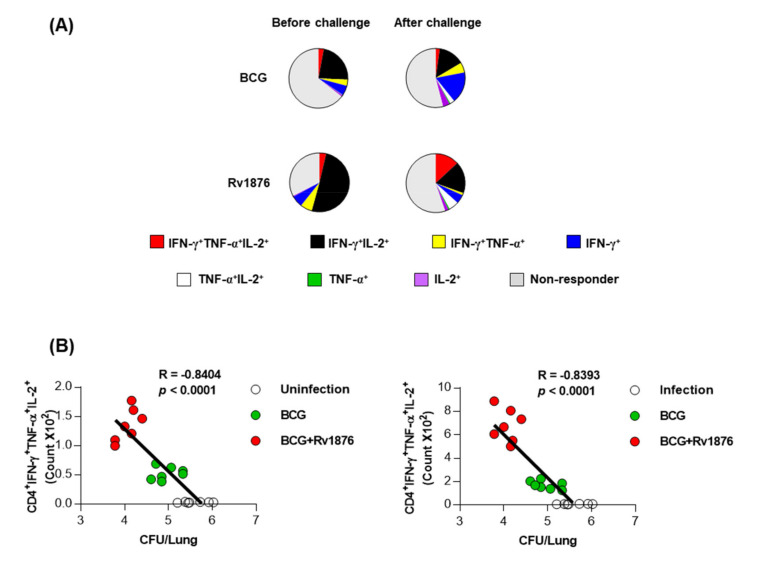
Induction of Ag-specific multifunctional T-cells in the lungs and analysis of protective correlations for protection levels. (**A**) Mice were immunized and euthanized as described in the Methods section. Four weeks after the last immunization, the mice were sacrificed, and the lung cells collected from the mice were treated with Rv1876 (2 μg/mL) at 37 °C for 12 h in the presence of GolgiStop. Upon stimulation with Rv1876, cell counts of Ag-specific, multifunctional CD4^+^CD44^+^ T-cells producing IFN-γ, and/or TNF-α and IL-2 in the lung cells from each immunized group were determined by flow cytometry. (**B**) The relationship between protection (CFU) and Rv1876 specific CD4^+^CD44^+^IFN-γ^+^TNF-α^+^IL-2^+^ producing T cells is shown as a fitted regression line with the correlation coefficient. Spearman’s *r* and *P* values of the correlations are indicated. White circle: naïve or infection, green circle: BCG, and red circle: BCG plus Rv1876.

## Data Availability

Data are available on request from the authors. The data that support the findings of this study are available from the corresponding author upon reasonable request.

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
