# Peer review of "A Dendritic Cell-Activating Rv1876 Protein Elicits Mycobacterium Bovis BCG-Prime Effect via Th1-Immune Response"

_biomolecules, 2021, doi:10.3390/biom11091306_

Round 1

Reviewer 1 Report

In this article, Choi et al identified a virulence-associated protein Rv1876 (bacterioferritin) in the Mycobacterium tuberculosis (Mtb) culture filtrate. In vitro, the authors showed that Rv1876 protein plays a critical role in the maturation of dendritic cells via MAPK and NFkb signaling, and in triggering Th1 immune responses, and in vivo,  it protected against challenge with Mtb in immunized mice with Rsv1876 adjuvanted with monophoryl lipid A, by expanding the effector/memory T cell subsets and reducing CFU counts in the lungs.

The authors suggest that Rv1876 could be promising in designing a novel vaccine to boost BCG.

Even though the paper is well written, I do not see any originality in this study. With this regard, the paper is almost similar from the experimental design point of you from one published by one author in 2012  (FASEB J 2012 Jun;26(6):2695-711. doi:10.1096/fj.11-199588), demonstrating that another virulence associated protein -Rv0577-  activated DCs and primes Th1 responses.  Moreover, in the reference list the authors either do not mention nor make any comment.

Perhaps it would be intersting to see if  boosting combinations of both protein would have a more strong effect.

Reviewer 2 Report

I read with great interest this high quality paper. Congratulations to the authors

Below my minor suggestions

Introduction: well the updata of TB data on global burden tb report 2020. I suggest only to write also on impact of COVID 19 pandemic on tb burden (see and cite Increase in Tuberculosis Diagnostic Delay during First Wave of the COVID-19 Pandemic: Data from an Italian Infectious Disease Referral Hospital. Antibiotics (Basel). 2021 Mar 8;10(3):272. )

Methods and results section: are perfect! very happy to read the paper

Discussion: add if you can the role of age/ immunosenesce and tb outcome (see Active Pulmonary Tuberculosis in Elderly Patients: A 2016-2019 Retrospective Analysis from an Italian Referral Hospital. Antibiotics (Basel). 2020 Aug 7;9(8):489.)

Conclusion: give some public health action and proposal (also future idea research) that came from your excellent paper

Reviewer 3 Report

The manuscript by Choi and co-workers describes the ability of the mycobacterial bacterioferritin, Rv1876, to induce in vitro dendritic cell maturation. In addition, using mice model, the authors present some interesting data indicating that Rv1876 could be used in a BCG-prime boost strategy. However, details are sometimes missing in the manuscript. Moreover, the figures 1 and 4 and their corresponding legends needs to be reworked in order to clarify the consistency of the work.

Lines 100 to 102: The construct of pET22b+rv1876 is a bit awkward as there is no EcoRI site in the second primer (but a HindIII site). Moreover, it should be specified that the sequence of the recombinant Rv1876 contains a C-terminal 6His-tag.

Lines 58-59, lines 93-96 and lines 231-232: The authors used different (self) references for the same protocol. Could it be possible to have a clear and brief description of the protocol used in the present study? Please also define the abbreviations HIC, HAT and IEC.

In direct relation with my previous point, Figure 1A is not clear at all.

The figure 1Aa does not show the activity of each fraction on DCs, as stated line 232.

What are pass fraction (HAT) and F5 (HIC)?

“Each fraction was verified for its ability to secrete pro-inflammatory cytokines from the immune cells” (lines 234-235). Where are the corresponding data?

“The primary fractions were divided and concentrated into seven fractions” (lanes 264-265). However, the gel displays eight lanes...

What is Figure 1876 (line 265)? What does the “p”” stand for in Rv1876p (Figure 1Ac)?

Legends for Figures 1Ab and 1Ac are missing.

In conclusion, please provide an understandable Figure 1A.

Line 240: How was assessed the endotoxin content?

Figure 1D: “Beta-actin was used as the loading controls for the cytosolic fractions” (line 284). However, the figure shows the alpha-tubulin content…

Figure 1F: Where is IL-10 graph (line 290)? What does “I” stand for? I guess this stands for non-treated by inhibitors.

Figure 3 (line 349): What is Table 85?

Figure 4A (line 381): What is Table 1876?

Figure 4A (line 383): What about the histopathological results?

Figure 4B (line 386): “n = 6 or 7 mice per group at each time point”. However, the graph displays only 4 dots per group…

Figure 4C: I guess black colour stands for BCG only and red colour stand for BCG+Rv1876 boost.

Line 469: Mb1907 is a M. bovis gene and not a BCG one. Maybe it would be more adequate to refer to Uniprot numbers, such as P9WPQ9 (for Rv1876) and A0A0K2HXD3 (for its equivalent in BCG strain). Anyways, it should be stated that their sequence is 100% identical.

Considering that BCG strains contain an identical protein that is probably produced considering its importance for mycobacterial physiology (see lines 418-426), what would the explanation for such differences between BCG only and BCG+Rv1876 treated cells? Is it only a dose-dependent effect?

Table 1 is incomplete for IL4.

Supplementary Figure 1: Stimulation of the Rv1876?

Supplementary Figure 2 : mDCs abbreviation is not defined, as well as in line 492 of the manuscript.

Round 2

Reviewer 1 Report

I am satisfied with the author's responses 

Reviewer 3 Report

Without a proper explanation, as provided in their cover letter (answer 20), the conclusion drawn from lines 498 and 499 is a bit clumsy.

Otherwise, the authors have addressed all my concerns.